# Premise-Augmented Reasoning Chains Improve Error Identification in Math Reasoning with LLMs

**Sagnik Mukherjee** [* 1]  **Abhinav Chinta** [* 1]  **Takyoung Kim** [1]  **Tarun Sharma** [1]  **Dilek Hakkani-Tür** [1]

## Abstract

Chain-of-Thought (CoT) prompting enhances mathematical reasoning in large language models (LLMs) by enabling detailed step-by-step solutions. However, due to the verbosity of LLMs, the resulting reasoning chains can be long, making it harder to verify the reasoning steps and trace issues resulting from dependencies between the steps that may be farther away in the sequence of steps. Importantly, mathematical reasoning allows each step to be derived from a small set of premises, which are a subset of the preceding steps in the reasoning chain. In this paper, we present a framework that identifies the premises for each step, to improve the evaluation of reasoning. We restructure conventional linear reasoning chains into **Premise Augmented Reasoning Chains (PARC)** by introducing premise links, resulting in a directed acyclic graph where the nodes are the steps and the edges are the premise links. Through experiments with a PARC-based dataset that we built, namely **PERL (Premises and ERrors identification in LLMs)**, we demonstrate that LLMs can reliably identify premises within complex reasoning chains. In particular, even open-source LLMs achieve 90% recall in premise identification. We also show that PARC helps to identify errors in reasoning chains more reliably. The accuracy of error identification improves by 6% to 16% absolute when step-by-step verification is carried out in PARC under the premises. Our findings highlight the utility of premise-centric representations in addressing complex problem-solving tasks and open new avenues for improving the reliability of LLM-based reasoning evaluations.

---

[*]Equal contribution  [1]University of Illinois at Urbana Champaign. Correspondence to: Sagnik Mukherjee <sagnikm3@illinois.edu>.

*Proceedings of the 42nd International Conference on Machine Learning*, Vancouver, Canada. PMLR 267, 2025. Copyright 2025 by the author(s).

## 1. Introduction

Chain-of-thought reasoning enhances problem solving by breaking down complex tasks into a series of logical steps, improving the accuracy and clarity of decision-making processes. There has been a well-documented success of the reasoning capabilities of LLMs with chain of thought reasoning (CoT; (Wei et al., 2023)) across applications such as embodied reasoning (Gao et al., 2023; Philipov et al., 2024), code generation (Jiang et al., 2024), and mathematical and scientific reasoning (Imani et al., 2023; Ahn et al., 2024). However, the corresponding evaluation methods and metrics have a significant limitation: they focus solely on the correctness of the *final* answer, neglecting intermediate reasoning processes and rationales that contribute to it. Although the final answer serves as a proxy for the reasoning capability, it is not sufficient to judge the reasoning performance of the model. Hence, evaluating the final answer provides a narrower view of the reasoning capabilities. Since the intermediate reasoning process is equally important as getting the correct final answer (Huang & Chang, 2023; Golovneva et al., 2023; Prasad et al., 2023), a comprehensive evaluation of the reasoning chains is crucial to holistically understanding the reasoning capabilities of LLMs.

Evaluation of reasoning chains has been studied in literature in the context of self-verification (Weng et al., 2023). Previous work has shown that LLMs struggle to find reasoning errors in chain-of-thought traces without the help of external verifiers (Stechly et al., 2024; Wu et al., 2024; Tyen et al., 2024), casting doubt on the overoptimism of LLMs' self-critique abilities. Existing research focusing on the evaluation of reasoning chains in LLMs can be broadly categorized into *reference-based* and *reference-free* methods. Reference-based methods, which rely on the availability of a ground truth reasoning chain (Welleck et al., 2021; Han et al., 2024; Tian et al., 2021), are reliable but are constrained by the significant cost of human annotations, restricting their application. In contrast, reference-free methods (Prasad et al., 2023; Golovneva et al., 2023; Zhu et al., 2024) bypass the need for annotations, but suffer from two major drawbacks: (1) most such works assign only a chain-level score rather than localizing specific errors, and (2) they often need fine-tuning of task-specific

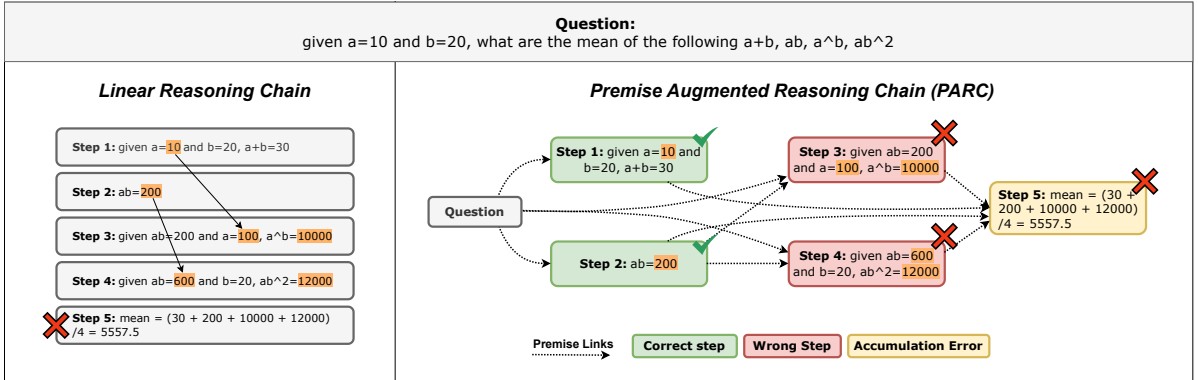

*Figure 1.* Comparison between a Linear Reasoning Chain (LRC) and our proposed PARC (Premise-Augmented Reasoning Chain). The LRC (left), is linear and there no explicit premise link between steps. In PARC (right), premise links are explicitly established, enabling better identification of correct and incorrect steps. Accumulation errors can be traced back to faulty premises. Establishing these premises helps improve error detection with LLMs.

models, restricting their generalizability. One workaround could be using formal proof assistants like Lean (Moura & Ullrich, 2021), which natively support the verifiability of generated proofs (Yang et al., 2024; 2023; Murphy et al., 2024), but this requires a challenging auto-formalization of natural language text (Wu et al., 2022) and often assumes the solution is already known for proof construction.

In our work, we focus on reference-free verification of LLM reasoning chains in the context of mathematical reasoning. Mathematical problem solving requires a series of deductive reasoning steps, where each step is performed under a small set of premises. We hypothesize that a step in a reasoning chain should be verified only under its premises. Previous work has shown that having unnecessary context hurts the performance of LLMs for solving math word problems (Shi et al., 2023). Thus, by excluding irrelevant context from the verifier, we make the verification of a reasoning step only conditioned on its premises, addressing the known susceptibility of LLMs to errors under distractors.

We restructure conventional linear reasoning chains (LRCs) into **PARC (Premise-Augmented Reasoning Chains)**, by identifying the premises of each step. Identifying those premises improves the traceability of a reasoning chain. We create a corresponding dataset, called **PERL (Premises and ERrors identification in Language models)** to test LLMs' capability in identifying premises as well as the effectiveness of premise augmentation in annotating both premises and errors. Additionally, we refine the error taxonomy (as proposed in Golovneva et al. (2023)) for mathematical reasoning by introducing a new category: "accumulation error." This error arises when a reasoning step is correct in isolation but is derived from flawed premises (refer to step 5 in Fig.1), resulting in error propagation throughout the chain; while such errors are common, existing work (Lightman et al., 2023; Zheng et al., 2024; Daheim et al., 2024) has not addressed them beyond discarding all subsequent steps

once the first mistake appears. Distinguishing accumulation errors from inherently flawed steps (mathematical errors or logical inconsistencies) is crucial for a holistic evaluation of CoT traces. Moreover, establishing explicit premise links between steps allows each step to be verified only under its premises which reduces irrelevant information for each verification step, thus improving the error detection for mathematical problem solving.

Our main contributions are as follows:

- We demonstrate that off-the-shelf LLMs can detect premises for a given step with high accuracy for mathematical reasoning, enabling the conversion of a Linear Reasoning Chain (LRC) to a Premise-Augmented Reasoning Chain (PARC).

- We establish that verifying each step under its corresponding premises increases the accuracy of identifying errors and their types.

- We propose a refined error taxonomy for mathematical reasoning, introducing the notion of *accumulation errors* for steps that are locally correct but inherit upstream errors.

- We introduce and will release **PERL**, a dataset of reasoning chains annotated with premises and error types, to facilitate broader research on premise-centered reasoning verification.

The rest of the paper is structured as follows: in Section 2, we discuss relevant prior work in context of mathematical reasoning and verification of reasoning chains with LLMs. Section 3 introduces the research questions, alongside the necessary background as well as mathematical definition of premises, conversion of LRC to PARC, as well as how we identify errors with premises. Section 4 details PERL, our dataset to show effectiveness of the framework, and lastly,

in section 5 we discuss the results and insights from our experiments. [1]

## 2. Related work

**Math Reasoning with LLMs.** Mathematical and scientific reasoning tasks have become a primary testbed to assess the capabilities of large language models (LLMs) (Hendrycks et al., 2021a; Arora et al., 2023; Wang et al., 2024). Solving these tasks requires complex planning, recall of relevant formulae, and grounding the solution to the given problem (Arora et al., 2023). A widely adopted approach to improve the reasoning of LLMs is to generate intermediate rationales, commonly known as chain-of-thought (CoT), before predicting a final answer (Wei et al., 2023). Recent work explores alternative structures for organizing these solutions, including Tree-of-Thoughts (Yao et al., 2023) and Graph-of-Thoughts (Besta et al., 2024).

**Evaluation of Chain-of-Thoughts.** Although generating rationales can significantly improve model reasoning, systematically evaluating these explanations remains a challenge. Traditional metrics focused on word overlap or embedding similarity (Celikyilmaz et al., 2021; Reiter, 2019) fail to capture logical soundness, especially for step-by-step deductions. To address this gap, methods such as ROSCOE (Golovneva et al., 2023), ReCEval (Prasad et al., 2023), and Socreval (He et al., 2024) offer reference-free evaluation frameworks that assess correctness and identify various categories of errors. However, these methods generally provide a single chain-level score and offer limited explainability, making it difficult to pinpoint specific error types in individual reasoning steps. In contrast, our proposed framework provides a natural language characterization of the correctness of each step, providing fine-grained insights into the precise nature of errors.

**Verifiers for Math Reasoning.** Verifiers have proven effective in enhancing LLM-based reasoning. Uesato et al. (2022) and Lightman et al. (2023) demonstrate that providing reward signals for intermediate steps can stabilize training and improve final performance. Deductive Beam Search (Zhu et al., 2024) similarly incorporates a trained verifier to refine model outputs during inference, leading to higher accuracy for the final task. While these studies highlight the benefits of verifiers, a clear strategy for training a robust verifier, one that generalizes across diverse problem formats, remains elusive. The closest existing effort, Ling et al. (2023), requires models to encode premises in a particular "natural program" style, which constrains applicability to general chains-of-thought. In contrast, our method treats

premise extraction as a standalone step and refrains from making strong assumptions about how the reasoning chain is structured.

## 3. Method

In this work, we explore how the establishment of premise links improves the identification of errors in reasoning chains with LLMs. Our approach works in two steps. First, we identify premises for each step and augment a linear reasoning chain (LRC) to convert it into a Premise Augmented Reasoning Chain (PARC). And then verify each step in the PARC under its premises only. Lastly, we do a graph traversal to identify steps that are logically correct but have faulty premises to identify accumulation errors. In particular, we are interested in the following research questions.

**RQ 1** Given a sequential step-by-step answer to a math word problem, can LLMs identify premises for each step?

**RQ 2** Given premise annotations, can LLMs identify errors in reasoning more faithfully?

**RQ 3** Can LLMs perform the entire process *end-to-end*, i.e., given a reasoning chain, can they identify premises for each step and detect errors?

### 3.1. Premise Augmented Reasoning Chains (PARC)

Let $\mathbf{q}$ represent the question, $\hat{a}$ the predicted answer, $a$ the ground truth answer, and $\mathbf{r}_{\leq \mathbf{t}} = [s_1, s_2, \ldots, s_t]$ the generated reasoning chain composed of $t$ intermediate steps $s_i$, leading to $\hat{a}$. Using CoT reasoning, the predicted answer $\hat{a}$ is derived by first generating a reasoning chain $\mathbf{r}_{\leq \mathbf{t}}$. The probability distribution for $\hat{a}$ can be expressed as:

$$\mathbb{P}_{\text{LM}}(\hat{a} \mid \mathbf{q}) = \mathbb{P}_{\text{LM}}(\hat{a} \mid \mathbf{r}_{\leq \mathbf{t}}) \times \mathbb{P}_{\text{LM}}(\mathbf{r}_{\leq \mathbf{t}} \mid \mathbf{q}),$$

where the reasoning chain $\mathbf{r}_{\leq \mathbf{t}}$ can be decomposed into intermediate steps as:

$$\mathbb{P}_{\text{LM}}(\mathbf{r}_{\leq \mathbf{t}} \mid \mathbf{q}) = \mathbb{P}_{\text{LM}}(s_1 \mid \mathbf{q}) \times \prod_{i=1}^{t-1} \mathbb{P}_{\text{LM}}(s_{i+1} \mid \mathbf{q}, s_i).$$

Our objective is to identify errors in the intermediate steps $s_i$ of the reasoning chain $\mathbf{r}_{\leq \mathbf{t}}$.

The *premises* for a step $\mathbf{s}_i$ are defined as the necessary and sufficient subset of prior steps, denoted as

$$\mathcal{P}_i \subseteq \{s_j \forall j < i\},$$

satisfying the following properties:

1. **Verifiability**: The correctness of $s_i$ is verifiable based on $\mathcal{P}_i$ alone:

$$\mathcal{F}(s_i \mid \mathcal{P}_i) = 1.$$

---

[1]Our code and data is available on https://github.com/SagnikMukherjee/PARC

2. **Minimality**: The set $\mathcal{P}_i$ is minimal such that removing any element $\mathbf{s}_j \in \mathcal{P}_i$ results in $s_i$ becoming unverifiable:

$$\mathcal{F}(s_i \mid \mathcal{P}_i \setminus \{s_j\}) = 0, \quad \forall s_j \in \mathcal{P}_i. \tag{1}$$

Given premises $\mathcal{P}_i$ for a step $\mathbf{s}_i$, we convert a linear reasoning chain into a PARC, where each step $\mathbf{s}_i$ is augmented with its premises $\mathcal{P}_i$. $\mathcal{F}$ is a function that estimates whether a step is verifiable or not.

$$\mathbf{r}_{\leq \mathbf{t}}' = [(s_1, \mathcal{P}_1), (s_2, \mathcal{P}_2), \ldots, (s_t, \mathcal{P}_t)],$$

where $\mathbf{r}_{\leq \mathbf{t}}'$ represents the transformed reasoning chain.

### 3.2. Progressive Premise Mapping

To transform a reasoning chain $\mathbf{r}_{\leq \mathbf{t}} = [s_1, s_2, \ldots, s_t]$ into a premise-augmented reasoning chain (PARC), each step $s_i$ must be explicitly linked to its premises $\mathcal{P}_i$. We explore two approaches to identify these premises: *Aggregative Premise Mapping* and *Dyadic Premise Mapping*.

#### 3.2.1. AGGREGATIVE PREMISE MAPPING

In the *Aggregative* approach, the premises for each step $\mathbf{s}_k$ are collectively identified by querying an LLM with the complete reasoning context up to the step. For a given question $\mathbf{q}$, the reasoning chain so far $\mathbf{r}_{<\mathbf{k}}$, and the next step $s_k$, the LLM is prompted to output $\mathcal{P}_k$.

#### 3.2.2. DYADIC PREMISE MAPPING

In the *Dyadic* approach, the task of identifying premises is reformulated as a pairwise evaluation. Instead of querying the LLM to identify all premises $\mathcal{P}_k$ for $s_k$ at once, we evaluate whether each individual step $s_i$ (where $i < k$) serves as a valid premise for $s_k$. For each pair $(s_i, s_k)$, the LLM is queried to compute:

$$\mathcal{I}(s_k \mid s_i) = \begin{cases} 1, & \text{if } s_i \text{ is a valid premise for } s_k, \\ 0, & \text{otherwise.} \end{cases}$$

The premises for $s_k$ are then given by:

$$\mathcal{P}_k = \{s_i \mid \mathcal{F}(s_k \mid s_i) = 1, \forall i < k\}.$$

### 3.3. Error Identification

Next, we present our setup for error identification and how premises play a significant role in this task. We introduce a taxonomy of error types that occur in math reasoning with LLMs and then describe our approach to identify them.

#### 3.3.1. TYPES OF ERROR

Traditionally, prior research has predominantly focused on identifying *native errors*, which refer to inaccuracies inherent to individual reasoning steps. These errors often arise

---

**Algorithm 1** Constructing and Evaluating PARC

**Input:** $R = [s_1, s_2, \ldots, s_t]$
// Step 1: Premise Extraction
**for** $k = 1$ **to** $t$ **do**
    $\mathcal{P}_k \leftarrow$ EXTRACTPREMISE$(s_k, \{s_1, \ldots, s_{k-1}\})$
**end for**
$R' \leftarrow [(s_1, \mathcal{P}_1), \ldots, (s_t, \mathcal{P}_t)]$
// Step 2: Error Detection
**for** $k = 1$ **to** $t$ **do**
    $\mathcal{E}_k^{(1)} \leftarrow$ ISMATHEMATICALERROR$(s_k)$
    $\mathcal{E}_k^{(2)} \leftarrow$ ISLOGICALLYINCONSISTENT$(s_k, \mathcal{P}_k)$
    $\mathcal{E}_k \leftarrow \mathcal{E}_k^{(1)} \vee \mathcal{E}_k^{(2)}$
**end for**
// Step 3: Accumulation Error Detection
**for** $k = 1$ **to** $t$ **do**
    **if** $s_k$ is correct **then**
        **for** $s_j \in \mathcal{P}_k$ **do**
            **if** $s_j$ is incorrect **then**
                $\mathcal{E}_k \leftarrow$ ACCUMULATIONERROR
                **break**
            **end if**
        **end for**
    **end if**
**end for**
$R' \leftarrow [(s_1, \mathcal{P}_1, \mathcal{E}_1), \ldots, (s_t, \mathcal{P}_t, \mathcal{E}_t)]$

---

from issues such as incorrect mathematical calculations (defined as *Mathematical Error*), logical irregularities (defined as *logical inconsistencies*) and are evaluated independently of the broader reasoning context. Although significant, this focus on native errors tends to overlook another critical category of errors, called *accumulation errors*.

An *accumulation error* occurs when a reasoning step is valid in isolation but is built upon one or more erroneous premises from earlier steps. Unlike native errors, accumulation errors emerge from the compounding effects of prior inaccuracies, propagating through sequential steps. Formally, in the context of the reasoning chain, a step $s_i$ is classified as:

1. *Correct*, if it contains no logical or mathematical errors, and is performed under premises that are also *correct*

2. A *native error* if it contains an inherent discrepancy (e.g., a miscalculation or logical inconsistency)

3. An *accumulation error* if $s_i$ is logically valid, but at least one of its premises is incorrect.

By addressing both native and accumulation errors, we can achieve a more comprehensive understanding of error dynamics in sequential reasoning processes.

#### 3.3.2. ERROR IDENTIFICATION

**Baseline.** In the baseline approach, a large language model (LLM) is used in a zero-shot setting to classify the reasoning steps. For a given question $\mathbf{q}$, the reasoning chain so far $r_{<k}$, and the next step proposed $s_k$, the model is

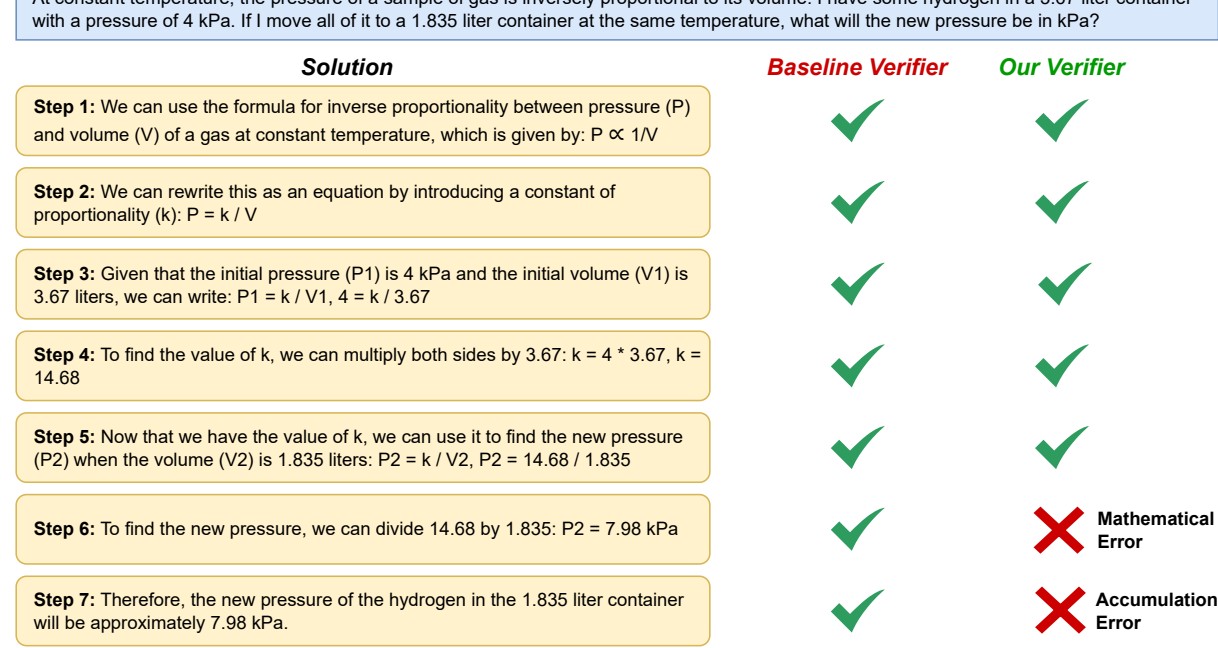

*Figure 2.* An example where the baseline method fails to detect errors, while our verification method with established premise links successfully identifies the mathematical error in step 6, and the accumulation error in step 7.

queried to assign an error type $\mathcal{E}_k$ for $s_k$ according to the predefined error taxonomy (*Correct / Mathematical Error / Logical Error / Accumulation Error*): The model processes the entire context $(\mathbf{q}, r_{<k}, s_k)$ and generates a classification directly. The zero-shot prompt explicitly defines all error types to guide the classification.

**Proposed Approach.** Our proposed approach refines the classification process by situating each reasoning step $\mathbf{s}_k$ explicitly within the context of its premises $\mathcal{P}_k$. Algorithm 1 illustrates the overall pipeline of conversion of LRC to PARC and step-by-step error classification. The prompts for the following are reported in Appendix 16 and 17.

**1. Mathematical Error:** To detect mathematical errors, we prompt an LLM to assess whether $s_k$ contains a mathematical error by analyzing the step in isolation, excluding the broader reasoning chain.

**2. Logical Inconsistency:** For the detection of logical inconsistencies, we restrict the evaluation to the premises $\mathcal{P}_k$ of the reasoning step $s_k$. The LLM is prompted to determine whether $s_k$ is logically consistent with its premises. This ensures that the step's validity is evaluated in the context of the dependencies defined by $\mathcal{P}_k$, without considering unrelated parts of the reasoning chain.

**3. Accumulation Error:** After identifying native errors, accumulation errors are identified by analyzing the dependency graph of the reasoning chain. The premises $\mathcal{P}_k$ form directed edges in the graph, where each step depends on its premises. A Depth-First Search (DFS) traversal is performed, and a step $s_k$ is classified as an Accumulation Error if it satisfies the following criteria: 1. $s_k$ itself is classified as correct, and 2. At least one premise $s_j \in \mathcal{P}_k$ (where $j < k$) is classified as incorrect.

## 4. PERL: Premises and ERrors identification in Language models

In order to test the ability of LLMs to identify premises and error categories, we designed a testbed. We used existing datasets of math word problems to create PERL, our testbed for step-level premise and error annotations.

**Generating Reasoning Chains:** In order to generate the reasoning chains, we used two popular benchmarks (i) GSM8K (Cobbe et al., 2021), a collection of 8,500 grade school math word problems, and (ii) MATH (Hendrycks et al., 2021b), a dataset of 12,500 challenging competition level math word problems. In addition, we use the (iii) Orca-Math dataset by Mitra et al. (2024), a synthetic dataset of 200K math problems alongside solutions written by GPT4Turbo, and the (iv) MetaMathQA dataset by (Yu et al., 2023). We first randomly sampled 1000 examples

| Model Name | GSM8K | | | MATH | | | Orca Math | | | MetaMathQA | | |
|---|---|---|---|---|---|---|---|---|---|---|---|---|
| | Precision | Recall | F1 | Precision | Recall | F1 | Precision | Recall | F1 | Precision | Recall | F1 |
| Llama 3.1 8b | 65.64 | 89.33 | 75.66 | 51.00 | 82.40 | 62.94 | 54.65 | 79.77 | 64.85 | 53.92 | 81.42 | 64.75 |
| Llama 3.1 70b | 81.26 | 97.55 | 88.65 | 69.80 | 96.82 | 81.11 | 71.05 | 96.73 | 81.91 | 73.10 | 96.46 | 83.05 |
| Qwen 2.5 7b | 67.42 | 63.53 | 65.36 | 55.87 | 59.73 | 57.70 | 63.08 | 72.64 | 67.05 | 55.54 | 61.66 | 58.14 |
| Qwen 2.5 32b | 84.07 | 95.35 | 89.34 | 72.56 | 88.55 | 79.74 | 74.27 | 90.63 | 81.62 | 75.57 | 90.89 | 82.49 |
| GPT4o-mini | 72.54 | 86.18 | 78.75 | 57.72 | 69.29 | 62.96 | 60.12 | 73.96 | 66.31 | 60.77 | 76.70 | 67.71 |
| GPT-4o | 85.93 | 94.42 | 89.96 | 69.40 | 89.78 | 78.28 | 71.12 | 92.17 | 80.79 | 75.49 | 90.37 | 81.85 |

*Table 1.* Precision, Recall and F1 scores for premise identification under the Aggregative approach. Note that ideally this needs to be a high recall system, because missing even a single premise step could hurt the verifiability of the reasoning chain.

| Model Name | Positives | | | Negatives | | | Synthetic Negatives | | |
|---|---|---|---|---|---|---|---|---|---|
| | Precision | Recall | F1 | Precision | Recall | F1 | Precision | Recall | F1 |
| **GPT4o-mini** | | | | | | | | | |
| Aggregative | 74.93 | 84.81 | 79.56 | 70.85 | 87.01 | 78.10 | 71.84 | 86.73 | 78.59 |
| Dyadic | 75.04 | 82.82 | 78.74 | 64.57 | 71.12 | 67.68 | 55.18 | 53.98 | 54.57 |
| **GPT4o** | | | | | | | | | |
| Aggregative | 89.38 | 93.66 | 91.47 | 84.97 | 94.95 | 89.69 | 83.43 | 94.65 | 88.72 |
| Dyadic | 75.32 | 99.01 | 85.55 | 67.70 | 92.68 | 78.25 | 67.27 | 79.42 | 72.84 |

*Table 2.* Precision, Recall and F1 scores for the premise mapping task for the GSM8K dataset under the aggregative and dyadic approaches. We saw a consistent drop in F1 score for the dyadic approach, in spite of it being computationally more expensive.

from the GSM8k and MATH test split and the Orca-Math and MetaMathQA training split (since these are training datasets). We began by using the open weight Large Language Model Llama-3.1-8B-Instruct (Grattafiori et al., 2024) to generate step-by-step reasoning chains. Next, we categorized the reasoning chains as positive or negative, depending on whether the final answer matches the ground-truth answer. Next, we randomly sampled 50 positive (correct) and 50 negative (incorrect) reasoning chains. To expand our dataset, we employed GPT-4o to systematically introduce mathematical or logical errors into the correct reasoning chains, creating additional synthetic negative examples (since prior work also focused on such synthetic negatives). When introducing errors, we ensured that the subsequent steps were appropriately modified to reflect the impact of the initial error. In contrast with existing works such as Golovneva et al. (2023), where the perturbation in a step is not followed up in subsequent steps, our synthetic negatives are more realistic.

**Premise and Error Annotation:** Next, for each step in the reasoning chain, we identified the ground truth premises for each step using the OpenAI o1-Preview model (OpenAI et al., 2024). Then, we use the same o1-Preview model to map each step to the predefined taxonomy of errors or mark it as correct, to create ground-truth error annotations. Upon obtaining the annotated data sets, 2 authors manually went through 10% of the generated data points to identify the discrepancy between the human annotations and the annotations done by the o1 model. Manual inspection of 71 samples showed that, in the case of premise annotations,

only 4 out of 71 had a discrepancy and 5 out of 71 had issues in error annotations. For premise identification, the discrepancy typically indicates additional steps as premises. However, for error annotation, the annotation errors do not have a clear pattern and manifest themselves in forms of misclassification of the correctness of the step. The prompts used to generate the annotations of the premises and errors are in Appendix A.3. A detailed summary of the dataset statistics can be found in Appendix A.2

## 5. Results and Discussion

For detailed information on our experimental setup, including model configurations and implementation details, refer to Appendix A.1.

### 5.1. Premise Mapping

In this section we try to answer RQ1 - "Can LLMs identify premises to convert a Linear Reasoning Chain to a PARC?"

**Metrics:** For identification of premises, we use precision, recall, and F1 score as metrics. However, we would want to highlight to the reader that the most important metric here is Recall. Since marking an extra step as a premise will not hurt the verifiability of the step, missing one will hurt its verifiability and all subsequent steps that rely on this step as a premise. Further, we note that some reasoning chains have a higher number of reasoning steps compared to others. In order to ensure robustness of the metrics, we first compute them at the level of each data point and later average the

metrics across the dataset. We tried two approaches for the task of premise mapping.

**Aggregative Approach:** Table 1 contains the results of the Aggregative approach for a suite of models and datasets. From the table, it is evident that LLMs are efficient in identifying premises for a given step. In particular, Llama 3.1 70b (Grattafiori et al., 2024), Qwen 2.5 32b (Qwen et al., 2025) and GPT4o are very efficient in this task, all having a recall greater than 90%. It also shows that these models can convert an LRC to a PARC with high accuracy.

We note that the scale of the model is an important factor. The increase in recall varies from 8.2% to 32.2% when we scale up for the same model class, and the pattern is consistent. Tables 9, 10, 12 and 11 in the appendix contain the detailed metrics across the splits Positive, Negative and Synthetic negatives for GSM8K, MATH, OrcaMath and MetaMathQA respectively.

**Dyadic Approach:** Table 2 provides a comparative analysis of precision, recall, and F1 scores for the GSM8K dataset using two models for the aggregative and dyadic approaches. This approach causes the LLM to be more biased to mark a step as a premise, causing precision to drop significantly. In addition, this approach has a time complexity of $O(n^2)$, where $n$ is the number of reasoning steps. Note that for almost all cases the precision drops, and the change in recall gives a mixed view on how this approach might be helpful.

Our analysis revealed that this methodology often led the model to incorrectly classify second- or higher-order dependencies as premises, even when the current step did not directly rely on them. This misclassification contributed to the observed degradation in precision.

### 5.2. Error Identification

In this section, we address RQ2 and RQ3, i.e. is error identification more robust when we only use the premises for a given step as context. We explain the experimental setup for error identification for the baseline, as well as our approach, in the previous sections. For the identification of errors under premises, we consider two settings. For the first one, we use the oracle premise annotation (from the ground truth). For the second, we first identify the premises with the LLMs and then use those premises for the identification of errors. Note that in the previous section we established that LLMs can identify premises with a high recall, and in this experiment, we exploit that. The prompts we used for our experiments are shared in Appendix A.5.

**Metrics:** For this task, we report the average accuracy for the identification of error types for each step. In our analysis, we observe that the boundaries between the error types *Mathematical Error* and *Logical Inconsistency* are

quite thin, and models often classify one as the other, so we merge these two error categories into a single error type *Error*, while keeping *Accumulation Error* separate. Furthermore, since an early erroneous step for a solution containing multiple steps could cause the number of errors to skew toward *accumulation errors*, we first normalize the accuracy by the number of steps in a datapoint, and later normalize across the dataset.

**Results:** Table 3 presents the experimental results for the error identification task. We evaluate performance under two setups: under complete context as input (*Full Context*), on the other hand, the *Model Premises* setup denotes the end-to-end approach, wherein the model is tasked with first identifying the relevant premises and subsequently utilizing them. All models are provided with the question and the solutions generated up to the current step and are tasked with predicting the error type observed in the current step.

**Premises Help Error Identification :** It is evident from Table 3, that providing only the premise steps as context (instead of providing the full context) improves the accuracy of error detection with LLMs. Figure 2 shows one such example, where the baseline approach (using an LRC, and error detection performed in the full context) could not identify errors, but our approach was able to successfully identify the mathematical and accumulation errors. Our results show that the models are not as biased towards marking a step as correct, when given a more precise context (no distractor). Hence, upon providing only the premises, identification of errors becomes much more robust. However, it is important to note that this also comes at the cost of a drop of accuracy for correct steps, this can be observed from comparing the Pos column for Full context and Model Premises. However, this accuracy drop is much smaller for larger models. In addition, our approach impacts larger and more capable models more than smaller models.

**Models Struggle With Error Detection, Especially Accumulation Errors:** We find that, while capable of identifying correct steps, all models struggle with identifying errors. This can be attributed to the fact that models are inherently biased toward marking a step as correct. This issue was also pointed out by Ling et al. (2023). Identifying errors with directly connected premises as context yields significantly better performance than simply providing the full context. Moreover, models struggle to identify accumulation errors. Table 5 contains the accuracies across error types in the negative split of GSM8K. Note that, in full context, accuracy is quite high for correct steps, but lower for erroneous steps (mathematical and logical errors), and significantly lower for accumulation errors. However, in our approach, the identification of errors becomes significantly robust, with a minor drop in performance in identifying correct steps.

| Model | GSM8K | | | | MATH | | | | Orca Math | | | | MetaMathQA | | | |
|---|---|---|---|---|---|---|---|---|---|---|---|---|---|---|---|---|
| | Neg | Syn | Pos | Avg | Neg | Syn | Pos | Avg | Neg | Syn | Pos | Avg | Neg | Syn | Pos | Avg |
| **Llama 3.1 8b** | | | | | | | | | | | | | | | | |
| Full Context | 48.87 | 57.68 | 91.9 | 58.6 | 46.91 | 60.29 | 90.61 | 60.2 | 43.26 | 51.2 | 96.7 | 55.06 | 50.47 | 55.15 | 95.17 | 59.45 |
| Model Premises | 54.61 | 76.74 | 65.44 | 64.53 | 52.16 | 59.83 | 50.96 | 54.88 | 55.57 | 59.82 | 67.36 | 59.22 | 54.52 | 69.13 | 62.63 | 61.78 |
| **Llama 3.1 70b** | | | | | | | | | | | | | | | | |
| Full Context | 58.35 | 77.03 | 98.61 | 71.37 | 55.55 | 72.83 | 96.24 | 69.77 | 50.33 | 64.25 | 96.74 | 63.52 | 55.59 | 73.13 | 97.93 | 69.49 |
| Model Premises | 74.51 | 86.22 | 94.69 | 81.92 | 70.94 | 79.29 | 77.91 | 75.45 | 64.95 | 72.28 | 93.37 | 72.52 | 65.69 | 82.74 | 89.71 | 76.52 |
| **Qwen 2.5 7b** | | | | | | | | | | | | | | | | |
| Full Context | 40.88 | 54.13 | 100 | 54.94 | 40.47 | 53.60 | 98.07 | 56.26 | 41.20 | 53.69 | 100 | 55.85 | 43.36 | 53.18 | 99.39 | 56.26 |
| Model Premises | 53.03 | 78.40 | 90.37 | 69.73 | 58.47 | 66.87 | 82.72 | 65.96 | 52.07 | 71.12 | 84.28 | 65.38 | 54.98 | 73.87 | 91.02 | 68.43 |
| **Qwen 2.5 32b** | | | | | | | | | | | | | | | | |
| Full Context | 46.76 | 69.26 | 99.8 | 63.56 | 49.58 | 66.5 | 98 | 65.11 | 49.44 | 63.01 | 99.5 | 63.06 | 49.7 | 67.8 | 99.6 | 65.02 |
| Model Premises | 65.58 | 86.34 | 96.95 | 79.37 | 68.61 | 76.96 | 88.71 | 75.57 | 61.89 | 69.24 | 95.25 | 70.26 | 62.18 | 85.60 | 97.32 | 77.40 |
| **Qwen 2.5 72b** | | | | | | | | | | | | | | | | |
| Full Context | 47.48 | 67.17 | 100 | 63.11 | 53.19 | 67.87 | 99.79 | 67.52 | 50.41 | 60.33 | 100 | 62.50 | 47.31 | 63.59 | 99.05 | 62.20 |
| Model Premises | 69.87 | 85.12 | 96.38 | 80.56 | 69.51 | 79.92 | 90.84 | 77.28 | 69.45 | 71.34 | 94.19 | 74.41 | 64.73 | 85.52 | 98.12 | 78.53 |
| **GPT4o-mini** | | | | | | | | | | | | | | | | |
| Full Context | 52.68 | 73.14 | 95.02 | 66.68 | 50.41 | 62.39 | 93.4 | 63.03 | 55.22 | 61.18 | 98.6 | 64.59 | 51.52 | 65.42 | 97.35 | 64.47 |
| Model Premises | 58.25 | 80.81 | 86.72 | 72.33 | 65.84 | 72.4 | 71.88 | 69.45 | 61.18 | 72.48 | 85.94 | 69.93 | 57.04 | 81.63 | 91.15 | 72.51 |
| **GPT-4o** | | | | | | | | | | | | | | | | |
| Full Context | 53.81 | 75.3 | 98.33 | 68.52 | 50.9 | 68.41 | 98.57 | 66.52 | 59.55 | 65.75 | 100 | 68.55 | 56.26 | 71.72 | 99.3 | 69.41 |
| Model Premises | 65.23 | 86.67 | 99.76 | 79.82 | 70.12 | 76.67 | 91.84 | 76.45 | 66.33 | 71.74 | 93.14 | 72.96 | 67.28 | 88.19 | 89.85 | 79.41 |

*Table 3.* Accuracy for Error identification of various models under Full Context (baseline) and Premises settings across different datasets. Note that Neg, Syn and Pos means Negative, Synthetic Negatives and Positive splits of PERL, as explained earlier. For each dataset, we highlighted the models that benefit the most from the PARC.

| Model | GSM8K | MATH | Orca-Math | MetaMath |
|---|---|---|---|---|
| **Llama 3.1 70b** | | | | |
| Oracle Premises | 81.46 | 74.68 | 73.45 | 76.05 |
| Model Premises | 81.92 | 75.45 | 72.52 | 76.52 |
| **Qwen 2.5 72b** | | | | |
| Oracle Premises | 80.58 | 79.50 | 74.87 | 79.49 |
| Model Premises | 80.56 | 77.28 | 74.41 | 78.53 |
| **GPT-4o** | | | | |
| Oracle Premises | 78.74 | 76.79 | 75.56 | 82.88 |
| Model Premises | 79.82 | 76.45 | 72.96 | 79.41 |

*Table 4.* Comparison of Error identification under oracle premises vs model generated premises. Since these models have high recall in premise identification, we observe that the error identification accuracy is comparable.

| Model | Correct | Error | Acc. Error | Avg |
|---|---|---|---|---|
| **Llama 3.1 70b** | | | | |
| Full Context | 96.79 | 60.87 | 12 | 58.35 |
| Oracle Premises | 90.18 | 75.25 | 55.63 | 74.41 |
| Model Premises | 88.78 | 75.25 | 57.54 | 74.51 |
| **GPT-4o** | | | | |
| Full Context | 95.75 | 48.2 | 13.2 | 53.81 |
| Oracle Premises | 96.83 | 60.02 | 41.79 | 67.03 |
| Model Premises | 94.15 | 55.13 | 44.91 | 65.23 |

*Table 5.* Error identification accuracy for each type of steps in ground truth. Acc. Error means Accumulation Errors

**Synthetic Negatives are Easier:** The average performance of all models on synthetically generated negatives is consistently higher compared to the performance on true negatives (evident from comparing the Neg and Syn columns in Table 3). This observation highlights an important insight: perturbing reasoning chains does not serve as a reliable proxy for evaluating a model's ability to identify genuine errors in reasoning. While prior work has focused on such synthetic negatives, it might provide an overoptimistic view of LLMs' ability to detect errors.

**Oracle Premises vs Model Generated Premises:** In Table 4, we present results of an ablation, where instead of providing the model generated premises, we provide the ground truth premises (oracle premises from PERL). As anticipated,

the accuracy of error identification improves when oracle premises are provided. However, it is noteworthy that for most models, the performance remains comparable to that achieved with oracle premises. This can be attributed to the fact that all these models can identify premises with a recall of higher than 90.

### 5.3. Experiments on ProcessBench

Beyond the PERL dataset, we also evaluate our framework on ProcessBench (Zheng et al., 2024), a benchmark that contains human-annotated stepwise labels for mathematical solutions. Unlike PERL, ProcessBench only includes annotations up to the *first* incorrect step in each solution.

**Experimental Setup.** We focus on the step-level error identification task, following the protocol used by Zheng et al. (2024). Specifically, for correct solutions (i.e., ones

| Model | Method | Correct | Wrong | F1 | Delta |
|---|---|---|---|---|---|
| **Qwen 2.5 7B** | Baseline | 66.3 | 36.7 | 47.2 | |
| | Ours | 60.1 | 38.6 | 47.0 | -0.2 |
| **Qwen 2.5 32B** | Baseline | 97.9 | 43.0 | 59.8 | |
| | Ours | 95.9 | 55.1 | 70.0 | +10.2 |
| **Qwen 2.5 72B** | Baseline | 98.4 | 61.4 | 75.6 | |
| | Ours | 97.8 | 59.7 | 74.1 | -1.5 |
| **Llama 3.1 8B** | Baseline | 17.1 | 36.7 | 23.3 | |
| | Ours | 33.7 | 37.8 | 35.6 | +12.3 |
| **Llama 3.1 70B** | Baseline | 77.7 | 57.5 | 66.1 | |
| | Ours | 89.6 | 70.0 | 78.6 | +12.5 |

*Table 6.* Performance comparison on the GSM8K split of Process-Bench with Baseline vs. PARC (Ours implies premise augmented verification) for various model.

| Model | Method | Correct | Wrong | F1 | Delta |
|---|---|---|---|---|---|
| **Qwen 2.5 7B** | Baseline | 46.0 | 25.4 | 32.7 | |
| | Ours | 45.6 | 41.2 | 43.3 | +10.6 |
| **Qwen 2.5 32B** | Baseline | 90.0 | 22.4 | 35.9 | |
| | Ours | 86.9 | 53.9 | 66.5 | +30.7 |
| **Qwen 2.5 72B** | Baseline | 88.5 | 33.7 | 48.8 | |
| | Ours | 86.7 | 53.9 | 66.5 | +17.7 |
| **Llama 3.1 8B** | Baseline | 5.6 | 19.1 | 8.7 | |
| | Ours | 11.0 | 27.5 | 15.7 | +7.1 |
| **Llama 3.1 70B** | Baseline | 32.4 | 32.8 | 32.6 | |
| | Ours | 61.6 | 55.4 | 58.3 | +25.7 |

*Table 7.* Performance comparison on the MATH split of Process-Bench with Baseline vs. PARC (Ours implies premise augmented verification) for various model.

where the final answer is correct), every step is marked as correct. For incorrect solutions, the task is to identify the *first* incorrect step in the chain; all subsequent steps are automatically deemed irrelevant by design of the benchmark. We adopt the same prompting strategy as in our previous experiments, with LLMs classifying each step as correct or incorrect. The baseline is a standard critic model as proposed by (Zheng et al., 2024) that identifies the first erroneous step from the solution as a whole.

**Results and Observations.** Tables 6 and 7 Our findings on ProcessBench (on the GSM8K and MATH splits) align with the trends we observe on PERL. *Premise-augmented verification* continues to outperform verification conducted in the full reasoning chain. We see consistent gains from evaluating each step under its extracted premises and classifying its correctness accordingly. This further strenghtens our previous findings.

## 6. Conclusion

In this paper, we introduced a framework to enhance the evaluation of mathematical reasoning chains in large language models by transforming Linear Reasoning Chains into Premise Augmented Reasoning Chains. Through exper-

iments with our annotated dataset, PERL, we empirically show that error identification under premises in PARC is more reliable and has higher accuracy than error identification under full context in LRC. We also show that LLMs can convert an LRC to PARC with no additional guidance, and then can do error identification under the premises in PARC. Further, we show that accumulation errors are particularly challenging to detect, and our method improves their identification by verifying each step under its premises.

## Acknowledgments

This research project has benefited from the Microsoft Accelerate Foundation Models Research (AFMR) grant program through which leading foundation models hosted by Microsoft Azure along with access to Azure credits were provided to conduct the research.

## Impact Statement

The broader impact of this research extends to multiple domains where step-by-step reasoning is critical, including automated theorem proving, educational AI tutors, and scientific discovery. By enabling more reliable self-verification mechanisms in LLMs, our approach contributes to the long-term goal of enhancing the transparency and trustworthiness of AI systems in high-stakes applications.

Potential ethical considerations include the risk of overreliance on LLM-generated verification without human oversight, particularly in domains where reasoning errors can have significant real-world consequences.

This paper primarily aims to advance the field of Machine Learning by proposing a structured approach to reasoning evaluation. Although we acknowledge that any advances in LLM-based reasoning may have broader societal implications, we do not foresee any immediate ethical concerns beyond those generally associated with the development of AI reasoning systems.

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

# A. Appendix

## A.1. Experimental Details

**Model**  For the Llama model (Grattafiori et al., 2024), we used vLLM (Kwon et al., 2023) for model serving and AzureOpenAI for the GPT4o and GPT4-o1 (OpenAI, 2024) models. To ensure reproducibility, all generations were performed with a temperature=0. For all models, we used their instruct variant.

## A.2. Data statistics

This results in a total of 607 reasoning chains with 203 positives, 214 negatives, and 190 synthetic negatives. In total, we have 2,134 steps annotated as *Correct*, 321 steps as *Mathematical Error*, 443 steps as *Logical Inconsistency*, and 741 steps as *Accumulation Error*, indicating that native and accumulation errors appear at almost equal rates. Additionally, we model each chain-of-thought as a directed acyclic graph (PARC) based on the step-level premise annotations. Across 607 PARCs, each chain contains on average 7.30 steps, representing the length of the chain-of-thought; 11.27 premises, corresponding to the total premise references across steps; and 10.42 edges linking premises to conclusions. The average depth of the PARC is 6.02, measuring the longest path of dependencies, while the maximum width is 1.90, quantifying how many steps can appear at the same layer in the PARC. Finally, the branching factor is 1.37, which is the ratio of edges to nodes, indicating that each step typically spawns fewer than two subsequent steps on average.

## A.3. Prompt for Error Annotation with O1

You are an expert mathematical reasoning analyzer. Your task is to analyze mathematical solutions and generate detailed error annotations in a specific JSON format. For each solution provided, you must carefully examine the reasoning chain and individual steps to identify any errors or issues.

Response Format

Your response must be a valid JSON object following exactly this structure:

```
{
  "error_annotations": {
    "chain_level": {
      "has_errors": boolean,
      "errors": [
        {
          "error_type": string,
          "error_description": string
        }
      ]
    },
    "step_level": [
```

```
      {
        "step_number": ,
        "has_error": boolean,
        "errors": [
          {
            "error_type": ,
            "error_description":
          }
        ]
      }
    ]
  }
}
```

Chain-Level Error Types

1. "Missing_Steps"
**Definition:** Solution lacks crucial concluding steps or final answer derivation
**Examples:**
- Not showing the final calculated value
- Missing the ultimate conclusion
- Failing to complete the proof
**Impact:** Makes the solution incomplete or inconclusive

2. "Planning_Error"
**Definition:** The reasoning takes an invalid or fundamentally incorrect approach
**Examples:**
- Using inapplicable theorems or methods
- Solving for incorrect variables
- Taking an approach that cannot possibly lead to a solution
**Impact:** Makes the entire solution path invalid
- Note: Valid but longer approaches (e.g., integration by parts instead of a substitution trick) should NOT be marked as errors

Step-Level Error Types

1. "Logical_Inconsistency"
**Definition:** Steps that violate logical principles or make unjustified conclusions
**Examples:**
- False equivalences
- Invalid deductions
- Unsupported assumptions
- Note that incorrect use of previous information (example the step uses a wrong value of a variable) is a Logical_Inconsistency
**Impact:** Breaks the logical flow of the solution

2. "Mathematical_Error"
**Definition:** Incorrect calculations, misuse of formulas, or mathematical operations
**Examples:**
- Arithmetic mistakes
- Incorrect algebraic manipulations

| Method | GSM8K | | | | MATH | | | | Orca Math | | | | MetaMathQA | | | |
|---|---|---|---|---|---|---|---|---|---|---|---|---|---|---|---|---|
| | Neg | Syn | Pos | Avg | Neg | Syn | Pos | Avg | Neg | Syn | Pos | Avg | Neg | Syn | Pos | Avg |
| *Full Context* | | | | | | | | | | | | | | | | |
| Llama 3.1 8b | 48.87 | 57.68 | 91.9 | 58.6 | 46.91 | 60.29 | 90.61 | 60.2 | 43.26 | 51.2 | 96.7 | 55.06 | 50.47 | 55.15 | 95.17 | 59.45 |
| Llama 3.1 70b | 58.35 | 77.03 | 98.61 | 71.37 | 55.55 | 72.83 | 96.24 | 69.77 | 50.33 | 64.25 | 96.74 | 63.52 | 55.59 | 73.13 | 97.93 | 69.49 |
| Qwen 2.5 7b | 40.88 | 54.13 | 100 | 54.94 | 40.47 | 53.60 | 98.07 | 56.26 | 41.20 | 53.69 | 100 | 55.85 | 43.36 | 53.18 | 99.39 | 56.26 |
| Qwen 2.5 32b | 46.76 | 69.26 | 99.8 | 63.56 | 49.58 | 66.5 | 98 | 65.11 | 49.44 | 63.01 | 99.5 | 63.06 | 49.7 | 67.8 | 99.6 | 65.02 |
| Qwen 2.5 72b | 47.48 | 67.17 | 100 | 63.11 | 53.19 | 67.87 | 99.79 | 67.52 | 50.41 | 60.33 | 100 | 62.50 | 47.31 | 63.59 | 99.05 | 62.20 |
| GPT4o-mini | 52.68 | 73.14 | 95.02 | 66.68 | 50.41 | 62.39 | 93.4 | 63.03 | 55.22 | 61.18 | 98.6 | 64.59 | 51.52 | 65.42 | 97.35 | 64.47 |
| GPT-4o | 53.81 | 75.3 | 98.33 | 68.52 | 50.9 | 68.41 | 98.57 | 66.52 | 59.55 | 65.75 | 100 | 68.55 | 56.26 | 71.72 | 99.3 | 69.41 |
| *Oracle Premises* | | | | | | | | | | | | | | | | |
| Llama 3.1 8b | 57.02 | 82.64 | 74.61 | 69.26 | 54.58 | 60.94 | 51.65 | 56.47 | 57.43 | 60.61 | 59.48 | 59.09 | 52.65 | 68.48 | 60.51 | 60.37 |
| Llama 3.1 70b | 74.41 | 85.12 | 94.69 | 81.46 | 68.55 | 79.88 | 78.11 | 74.68 | 66.78 | 72.65 | 93.37 | 73.45 | 63.31 | 83.15 | 92.16 | 76.05 |
| Qwen 2.5 7b | 52.40 | 79.37 | 89.56 | 69.74 | 56.86 | 70.23 | 79.85 | 65.96 | 55.01 | 66.66 | 76.65 | 63.39 | 56.32 | 77.06 | 91.94 | 70.45 |
| Qwen 32b | 66.57 | 88.64 | 96.95 | 80.76 | 69.24 | 82.26 | 89.32 | 77.98 | 63.58 | 69.88 | 96.31 | 71.4 | 59.84 | 85.74 | 96.87 | 76.38 |
| Qwen 2.5 72b | 69.48 | 86.34 | 98.98 | 80.58 | 72.98 | 81.10 | 91.03 | 79.50 | 69.76 | 72.77 | 93.24 | 74.87 | 66.23 | 86.21 | 98.06 | 79.50 |
| GPT4o-mini | 63.83 | 85.77 | 91.29 | 76.16 | 65.86 | 74.99 | 78.41 | 71.71 | 64.03 | 72.66 | 83.56 | 70.74 | 58.77 | 85.11 | 91.44 | 74.73 |
| GPT-4o | 67.03 | 87.2 | 94.76 | 78.74 | 70.32 | 77.3 | 90.65 | 76.79 | 71.19 | 73.69 | 92.15 | 75.56 | 72.76 | 88.95 | 94.82 | 82.88 |
| *Model Premises* | | | | | | | | | | | | | | | | |
| Llama 3.1 8b | 54.61 | 76.74 | 65.44 | 64.53 | 52.16 | 59.83 | 50.96 | 54.88 | 55.57 | 59.82 | 67.36 | 59.22 | 54.52 | 69.13 | 62.63 | 61.78 |
| Llama 3.1 70b | 74.51 | 86.22 | 94.69 | 81.92 | 70.94 | 79.29 | 77.91 | 75.45 | 64.95 | 72.28 | 93.37 | 72.52 | 65.69 | 82.74 | 89.71 | 76.52 |
| Qwen 2.5 7b | 53.03 | 78.40 | 90.37 | 69.73 | 58.47 | 66.87 | 82.72 | 65.96 | 52.07 | 71.12 | 84.28 | 65.38 | 54.98 | 73.87 | 91.02 | 68.43 |
| Qwen 32b | 65.58 | 86.34 | 96.95 | 79.37 | 68.61 | 76.96 | 88.71 | 75.57 | 61.89 | 69.24 | 95.25 | 70.26 | 62.18 | 85.60 | 97.32 | 77.40 |
| Qwen 2.5 72b | 69.87 | 85.12 | 96.38 | 80.56 | 69.51 | 79.92 | 90.84 | 77.28 | 69.45 | 71.34 | 94.19 | 74.41 | 64.73 | 85.52 | 98.12 | 78.53 |
| GPT4o-mini | 58.25 | 80.81 | 86.72 | 72.33 | 65.84 | 72.4 | 71.88 | 69.45 | 61.18 | 72.48 | 85.94 | 69.93 | 57.04 | 81.63 | 91.15 | 72.51 |
| GPT-4o | 65.23 | 86.67 | 99.76 | 79.82 | 70.12 | 76.67 | 91.84 | 76.45 | 66.33 | 71.74 | 93.14 | 72.96 | 67.28 | 88.19 | 89.85 | 79.41 |

*Table 8.* Detailed results for error identification for all dataset and models

| Model Name | Positives | | | Negatives | | | Synthetic Negatives | | |
|---|---|---|---|---|---|---|---|---|---|
| | Precision | Recall | F1 | Precision | Recall | F1 | Precision | Recall | F1 |
| Llama 3.1 8b | 67.80 | 89.91 | 77.31 | 65.03 | 91.31 | 75.96 | 64.08 | 86.76 | 73.71 |
| Llama 3.1 70b | 83.49 | 97.09 | 89.78 | 80.36 | 98.87 | 88.66 | 79.92 | 96.68 | 87.51 |
| Qwen 7b | 69.94 | 62.85 | 66.21 | 66.14 | 67.60 | 66.86 | 66.18 | 60.15 | 63.02 |
| Qwen 32b | 85.98 | 93.49 | 89.58 | 83.78 | 96.95 | 89.88 | 82.45 | 95.61 | 88.54 |
| GPT4o-mini | 74.93 | 84.81 | 79.56 | 70.85 | 87.01 | 78.10 | 71.84 | 86.73 | 78.59 |
| GPT-4o | 89.38 | 93.66 | 91.47 | 84.97 | 94.95 | 89.69 | 83.43 | 94.65 | 88.72 |

*Table 9.* Precision, Recall and F1 score for premise identification with GSM8K

| Model Name | Positives | | | Negatives | | | Synthetic Negatives | | |
|---|---|---|---|---|---|---|---|---|---|
| | Precision | Recall | F1 | Precision | Recall | F1 | Precision | Recall | F1 |
| Llama 3.1 8b | 53.98 | 82.36 | 65.22 | 46.39 | 82.43 | 59.37 | 52.63 | 82.42 | 64.24 |
| Llama 3.1 70b | 71.13 | 95.82 | 81.65 | 67.82 | 97.16 | 79.88 | 70.46 | 97.47 | 81.80 |
| Qwen 7b | 59.40 | 60/30 | 59.85 | 48.82 | 56.02 | 52.17 | 59.40 | 62.87 | 61.09 |
| Qwen 32b | 74.57 | 87.30 | 80.43 | 69.98 | 87.31 | 77.69 | 73.13 | 91.04 | 81.11 |
| GPT4o-mini | 57.58 | 66.57 | 61.75 | 54.42 | 69.01 | 60.85 | 61.18 | 72.29 | 66.27 |
| GPT-4o | 69.76 | 90.19 | 78.67 | 66.95 | 87.16 | 75.73 | 71.48 | 91.99 | 80.45 |

*Table 10.* Precision, Recall and F1 score for premise identification in MATH

- Wrong formula application
- Note that Mathematical_Error can only appear when there is an error in calculation
**Impact:** Produces incorrect numerical or algebraic results

3. "Accumulation_Error"
**Definition:** Errors that propagate from previous incorrect steps

**Examples:**
- Using wrong intermediate results
- Building upon previously miscalculated values
**Impact:** Compounds previous mistakes into larger errors

4. "Other"
**Definition:** Any error that doesn't fit into the above cate-

| Model Name | Positives | | | Negatives | | | Synthetic Negatives | | |
|---|---|---|---|---|---|---|---|---|---|
| | Precision | Recall | F1 | Precision | Recall | F1 | Precision | Recall | F1 |
| Llama 3.1 8b | 58.94 | 84.52 | 69.45 | 44.14 | 76.76 | 56.05 | 58.67 | 82.99 | 68.74 |
| Llama 3.1 70b | 77.82 | 97.37 | 86.50 | 65.18 | 96.13 | 77.69 | 76.29 | 95.88 | 84.97 |
| Qwen 7b | 58.05 | 59.09 | 58.57 | 48.72 | 66.48 | 56.23 | 59.86 | 59.41 | 59.63 |
| Qwen 32b | 78.73 | 91.60 | 84.68 | 69.89 | 88.99 | 78.29 | 78.09 | 92.07 | 84.50 |
| GPT4o-mini | 65.06 | 76.92 | 70.50 | 51.74 | 73.81 | 60.84 | 65.52 | 79.37 | 71.78 |
| GPT-4o | 79.91 | 92.51 | 84.68 | 67.13 | 88.11 | 76.26 | 79.42 | 90.49 | 84.62 |

*Table 11.* Precision, Recall and F1 score for premise identification with MetaMathQA.

| Model Name | Positives | | | Negatives | | | Synthetic Negatives | | |
|---|---|---|---|---|---|---|---|---|---|
| | Precision | Recall | F1 | Precision | Recall | F1 | Precision | Recall | F1 |
| Llama 3.1 8b | 56.02 | 80.12 | 65.94 | 52.60 | 78.92 | 63.13 | 55.31 | 80.26 | 65.49 |
| Llama 3.1 70b | 72.82 | 96.83 | 83.13 | 68.15 | 96.42 | 79.86 | 72.17 | 96.93 | 82.74 |
| Qwen 7b | 62.33 | 61.81 | 62.07 | 68.15 | 96.42 | 79.86 | 58.75 | 59.69 | 59.22 |
| Qwen 32b | 77.73 | 91.18 | 83.92 | 71.55 | 89.55 | 79.54 | 73.55 | 91.16 | 81.41 |
| GPT4o-mini | 63.24 | 75.36 | 68.77 | 57.36 | 73.38 | 64.39 | 59.75 | 73.13 | 65.77 |
| GPT-4o | 73.90 | 92.63 | 83.92 | 66.97 | 92.87 | 77.92 | 72.50 | 91.02 | 80.53 |

*Table 12.* Precision, Recall and F1 score for premise identification in OrcaMath

gories Examples:
- Notation mistakes
- Unclear explanations
- Formatting issues
**Impact:** Varies depending on the specific error

Analysis Requirements
1. Examine each step against mathematical principles and theorems
2. Verify all calculations and mathematical operations
3. Check for proper use of definitions and formulas
4. Ensure logical flow between steps
5. Compare against the provided ground truth answer
6. Consider the completeness of the solution

Important Notes
- Provide ONLY the JSON output, no additional text or explanations
- Every step in the solution must have a corresponding entry in step_level array
- Keep error descriptions clear, specific, and mathematically precise
- Use empty arrays for errors when no errors exist
- Ensure your response is always valid JSON that matches the exact format specified
- Each error must have both an error_type and a corresponding detailed error_description
- Error descriptions should be specific to the mathematical context of the problem
- Do NOT penalize valid but verbose approaches (e.g., breaking down algebra into multiple steps)
- Do NOT mark alternative solution methods as errors unless they are genuinely invalid

- Focus on correctness rather than elegance or brevity

Workflow
1. Read and understand the problem statement
2. Analyze the reasoning chain step-by-step
3. Check for chain-level errors
4. Analyze each step for specific errors
5. Verify all premises and justifications
6. Ensure completeness of the solution

### A.4. Prompt for Premise Annotation with O1

The system prompt and the instruction for the O1 model for identifying premises for a given step is shared in Table 13

### A.5. Prompt for Error identification

The prompts used for the baseline approach are shared in Appendix 14 and 15 . The evaluation for our error identification with premises are done with the prompts outlined in Tables 16 and 17,

**Instruction:**
Given this math word problem and its solution steps, identify the key premises and their relationships.
**Problem:** {problem['question']}
**Solution Steps:**
{chr(10).join(problem['steps'])}
**Return your analysis in this exact JSON format:**
{json_template}
**Critical Rules for Premises:**
1. A step can NEVER use itself as a premise. For example, Step 3 cannot use any premise labeled as [3, "..."].
2. Premises can only come from:
- Step 0 (problem statement and fundamental mathematical principles)
- Previous steps (steps with lower index)
3. Any intermediate calculations or logical steps within a single step should be part of that step's reasoning, not treated as separate premises.
4. Mathematical principles (like properties of operations) should be treated as part of Step 0.

**Additional Requirements:**
1. Start with Step 0 containing the problem statement.
2. For each step after 0, copy the EXACT text from the student's solution into 'original_step'.
3. Show clear reasoning for how premises lead to conclusions.
4. Return ONLY valid JSON with no additional text.
5. Do not use any special characters like &, ¡, ¿, etc.
6. Do not add any additional text for formatting it (e.g., "json"), just output the raw JSON.
7. Maintain the exact same number of steps as in the original solution.

**Remember:**
- Each step's premises must strictly come from either the problem statement (Step 0) or previous steps. Never from the current step.
- Keep each step atomic—do not split steps into multiple substeps even if they contain multiple calculations.
- The number of steps in your output (excluding Step 0) must match exactly with the number of steps in the student's solution.

**System Prompt:**
You are an expert in mathematical reasoning. Your task is to analyze solution steps and output a JSON object containing:
1. The premises used in each step.
2. The conclusion reached.
3. The reasoning that connects premises to conclusions.

Output MUST be valid JSON with no additional text or explanation.

*Table 13.* System prompt and instruction for the O1 model to identify premises for a given step

**Instruction:**
Question: {question}
Solution so far: {solution}

1. "Logical_Inconsistency"
- **Definition**: Steps that violate logical principles or make unjustified conclusions
- **Examples**:
- False equivalences
- Invalid deductions
- Unsupported assumptions
- Note that incorrect use of previous information (example the step uses a wrong value of a variable) is a Logical_Inconsistency
- **Impact**: Breaks the logical flow of the solution

2. "Mathematical_Error"
- **Definition**: Incorrect calculations, misuse of formulas, or mathematical operations
- **Examples**:
- Arithmetic mistakes
- Incorrect algebraic manipulations
- Wrong formula application
- Note that Mathematical_Error can only appear when there is an error in calculation
- **Impact**: Produces incorrect numerical or algebraic results

3. "Accumulation_Error"
- **Definition**: Errors that propagate from previous incorrect steps
- **Examples**:
- Using wrong intermediate results
- Building upon previously miscalculated values
- **Impact**: Compounds previous mistakes into larger errors

4. "Other"
- **Definition**: Any error that doesn't fit into the above categories
- **Examples**:
- Notation mistakes
- Unclear explanations
- Formatting issues
- **Impact**: Varies depending on the specific error

Statement to analyze: step
Format your response as: Reasoning: [detailed analysis of the statement's validity] Verdict: [CORRECT, Mathematical_Error, Logical_Inconsistency, or Accumulation_Error]

*Table 14.* Baseline error identification Instruction

---

**System Prompt:**
You are a helpful AI assistant that analyzes mathematical solution steps. Your task is to determine if each statement is COMPLETELY correct by carefully analyzing its validity. Focus ONLY on whether the current step is valid - do not consider whether it helps reach the final answer or whether better steps could have been taken. Mark a statement as CORRECT unless you find a specific error.

---

*Table 15.* Baseline error identification System prompt

---

**System Prompt:**
Your task is to determine whether a given sentence contains any mathematical errors. For mathematical error, check if the sentence contains mathematical calculations (arithmetic or algebraic), and whether they are incorrect. If there are such errors, mark the sentence as "mathematical_error" - Mathematical errors can only come from incorrect results of mathematical operations If no such errors exist, mark it as "correct".
Note: mathematical error can only come from incorrect numerical or algebraic calculations (i.e. wrong multiplication, wrong addition etc.) if there are no numerical or algebraic calculations done, you can mark it as correct
**Instruction :**
Statement to analyze:
step
Format your response as:
Reasoning: [detailed analysis of the statement's validity]
Verdict: [correct or mathematical_error]

---

*Table 16.* Math error identification instruction

---

**System Prompt:**
You are provided with a math question, a statement which is a step in the solution to the question and the premises to this steps (the question is also a premise). Your task is to identify whether the step follow naturally from the premises or not. If the current step contradicts the premises, mark is as a logical_inconsistency If the step can be directly inferred from the premises, mark it as correct. You should not check whether the premises are correct, assume they are correct. Only check the sentence given.
**Instruction :**
Given Premises:
Question: {question}
Previous steps as premise: {premises}
Statement to analyze: {step}
Guidelines:
1. for logical_inconsistency check if the step was performed under misinterpretation of the premises, made invalid deductions or had unsupported assumptions
2. Don't check for correctness of the premises, your only task is to check correctness of the given sentence
Format your response as: Reasoning: [detailed analysis of the statement's validity] Verdict: [correct, logical_inconsistency]

---

*Table 17.* Logical error identification instruction

**Instruction:**
You are provided with a question, a partial solution, and the next step in the solution.
Your task is to identify the steps that serve as premises for the given next step.
A step qualifies as a premise if the next step directly relies on information from that step. Based on the identified premises, the correctness of the next step should be fully verifiable.
Question (Step 0):
{question}
Solution so far:
{solution}
Next step to analyze:
{step}
For the step above, identify which previous steps (including Step 0 - the question) are premises and explain why each one is necessary. Remember:
1. A step cannot be a premise to itself
2. The question (Step 0) can be a premise if used directly
Generate ONLY the premises and nothing else.
Format your response with one premise per line as:
Step X: [explanation of why this step is necessary for the current step]

*Table 18.* Prompt for evaluating models in the premise identification task (zero shot)

