# OpenReview forum: "Premise-Augmented Reasoning Chains Improve Error Identification in Math reasoning with LLMs"
_ICML.cc/2025/Conference — ICML 2025 poster_

### Official Review · Reviewer_kjS7 · 2025-03-10

**Overall Recommendation:** 3

**Summary:**

This paper looks into the problem of reference-free verification of LLM reasoning chains in the context of mathematical reasoning. Authors hypothesize that a step in a reasoning chain should be verified only under its premises, and propose constructing Premise-Augmented Reasoning Chains (PARC) to improve the traceability of a reasoning chain. Authors create a corresponding dataset, called PERL (Premises and ERrors identification in Language models) to test LLMs’ capability in identifying premises as well as the effectiveness of premises.

**Claims And Evidence:**

Through extensive experiments on Math datasets, authors show that verifying each step under its corresponding premises increases the accuracy of identifying errors and their types. In their error design, authors consider 3 error types specific to mathematical reasoning: mathematical error, logical inconsistencies, and accumulation errors. Authors highlight the importance of newly introduced accumulation errors, but do not provide evidence in their study (i.e. through the ablation study).

**Essential References Not Discussed:**

N/A

**Experimental Designs Or Analyses:**

Study is limited to the four popular mathematical datasets, and three commonly used LLM. Results are consistent across experiments.

**Methods And Evaluation Criteria:**

yes

**Other Comments Or Suggestions:**

N/A

**Other Strengths And Weaknesses:**

Overall, paper is clear and well written. As LLMs are often used as Judges and Verifiers, it is important to know how good they actually are at this task, how can we meta-evaluate their evaluation skills, and improve on them. Proposed approach can be used to enhance  evaluations of the mathematical reasoning. In my opinion, this paper could benefit from additional ablation study.

**Questions For Authors:**

1.  You compare with baseline that classifies each reasoning step according to the predefined error taxonomy. In reality, there can be other error types outside those you pre-defined. I have noticed you did have "Other" error type in the baseline prompt (Table 12, p18). Do you have any statistics about the final error distribution within each dataset in the baseline setup? how large is the "other" group, and what's inside?
2. Paper could benefit from additional ablation study that would highlight the importance of the proposed taxonomy. In particular, it would be interesting to see how each aspect - mathematical error, logical inconsistencies, and accumulation errors - affects final error detection.

**Relation To Broader Scientific Literature:**

Authors elaborate on existing research, generalizing previously developed error taxonomy (i.e. Roscoe), and extensive work on using Verifiers to improve model's reasoning abilities.

**Theoretical Claims:**

N/A

---

> ### Author Rebuttal · Authors · 2025-04-01
>
> We thank the reviewer for their kind remarks on the writing and experimental details of the paper. Here we address their concerns
>
> **concern 1** - Study is limited to the four popular mathematical datasets, and three commonly used LLM
>
> **response** - Please note that we used three popular model families, and for each family we included at least 2 models of different scales, making the number of models=6. Which establishes the robustness of our method across models. Further here we attach some additional experiments done on the recently released processbench dataset for the error identification task, where across model classes we see significant improvements in error identification, further showcasing our method’s reliability
>
> Gsm8k
> | Model      | Method   | Correct acc | Wrong acc | F1   | Δ (Delta) |
> |------------|----------|-------------|-----------|------|-----------|
> | Qwen 7B    | Baseline | 66.3        | 36.7      | 47.2 | –         |
> |            | Ours     | 60.1        | 38.6      | 47.0 | -0.2      |
> | Qwen 72B   | Baseline | 98.4        | 61.4      | 75.6 | –         |
> |            | Ours     | 97.8        | 59.7      | 74.1 | -1.5      |
> | Qwen 32B   | Baseline | 97.9        | 43.0      | 59.8 | –         |
> |            | Ours     | 95.9        | 55.1      | 70.0 | 10.2      |
> | Llama 8B   | Baseline | 17.1        | 36.7      | 23.3 | –         |
> |            | Ours     | 33.7        | 37.8      | 35.6 | 12.3      |
> | Llama 70B  | Baseline | 77.7        | 57.5      | 66.1 | –         |
> |            | Ours     | 89.6        | 70.0      | 78.6 | 12.5      |
>
>
> Math
>
> | Model      | Method   | Correct acc | Wrong acc | F1   | Δ (Delta) |
> |------------|----------|-------------|-----------|------|-----------|
> | Qwen 7B    | Baseline | 46.0        | 25.4      | 32.7 | –         |
> |            | Ours     | 45.6        | 41.2      | 43.3 | 10.6      |
> | Qwen 72B   | Baseline | 88.5        | 33.7      | 48.8 | –         |
> |            | Ours     | 86.7        | 53.9      | 66.5 | 17.7      |
> | Qwen 32B   | Baseline | 90.0        | 22.4      | 35.9 | –         |
> |            | Ours     | 86.9        | 53.9      | 66.5 | 30.7      |
> | Llama 8B   | Baseline | 5.6         | 19.1      | 8.7  | –         |
> |            | Ours     | 11.0        | 27.5      | 15.7 | 7.0       |
> | Llama 70B  | Baseline | 32.4        | 32.8      | 32.6 | –         |
> |            | Ours     | 61.6        | 55.4      | 58.3 | 25.7      |
>
> **concern 2** - how large is the "other" group, and what's inside
>
> **response** - We observed that the frequency of the other error type in our ground truth dataset was quite low, where only 10 steps were marked as Other. Upon manual inspection, it revealed that the other group is manifested in the forms of - “The solution becomes stuck in a repetitive loop” and “Step is incomplete ”.
>
> **concern 3** - In particular, it would be interesting to see how each aspect - mathematical error, logical inconsistencies, and accumulation errors - affects final error detection
>
> **response** - Since we are restricted by 5000 characters, we request the reviewer to kindly see the response to the third concern by reviewer xai3, where we shared the detailed numbers.
>
> **concern 4** - Authors highlight the importance of newly introduced accumulation errors, but do not provide evidence in their study
>
> **response** - A holistic evaluation of LLM reasoning should consider the entire reasoning chain rather than relying solely on a binary correct/incorrect outcome of the final answer. Reasoning chains may contain subtle intermediate errors despite following a globally correct plan, ultimately rendering the solution incorrect—a nuance overlooked by final-answer correctness metrics. Prior works like PRM800K and ProcessBench have typically annotated reasoning chains only up to the first erroneous step, discarding subsequent steps due to ambiguity. To our knowledge, we are the first to formally introduce the concept of accumulation errors, enabling a more comprehensive evaluation of reasoning chains. Similar to how teachers award partial credit for nearly correct answers, evaluation frameworks should recognize when the overall reasoning plan is sound despite minor mistakes, assigning partial credit accordingly. Accumulation errors, where a step is locally correct but built on flawed premises, explicitly capture this subtlety. Identifying accumulation errors highlights how earlier mistakes compromise the reliability of the reasoning chain, making it essential to incorporate these errors into holistic scoring methods.

---

### Official Review · Reviewer_K1LJ · 2025-03-11

**Overall Recommendation:** 3

**Summary:**

The authors explore how to improve error identification in reasoning chains, which consist of multiple individual reasoning steps. They start by converting the reasoning chain into a directed acyclic graph, called Premise-Augmented Reasoning Chains (PARC), where the nodes that are reasoning steps and the edges indicate the dependency on premises from previous reasoning steps. They present two LLM-based approaches: providing the full reasoning chain until the current step and asking the LLM to identify the premises of the current step and alternatively building pairs of the current step with each of its predecessors and asking the LLM, whether a given step is the premise of the current step. Additionally to mathematically errors within a reasoning step and logical inconsistencies, where a reasoning step is not consistent with its premises, the authors also propose a new error type: the accumulation error, where the reasoning itself is correct, but depends on incorrect premises. The authors also derive a dataset, called Premises and ERrors identification in Language models (PERC), which they intend to publish, that is based on 607 positive, negative and synthetic negative samples, which they subsequent use to evaluate seven different models. They find that LLMs can identify the premises reasonably well and that their approach improves error identification.

## update after rebuttal

While the rebuttals answered some of my questions, the addition of another dataset has not convinced me enough to raise my score.

**Claims And Evidence:**

The claims are reasonably support by their evaluation.

**Essential References Not Discussed:**

None to my knowledge.

**Experimental Designs Or Analyses:**

The experiments are reasonably well designed, albeit follow their approach very much. The author provide some end-to-end results, but only for their own dataset. The authors supply some form of ablation study, where they test different approaches.

* If my understanding is correct, the baseline performance of these models is missing, making it harder to judge by how much the whole approach improves upon the base models.
* No discussion of the additional compute resources, time and costs.

**Methods And Evaluation Criteria:**

I think the methodology is sound in general and the authors evaluate seven models, showing that their approach at least generalizes in this dimension. The chosen metrics seem appropriate.

* The benchmark seems a bit small: only 607 datapoints.
* No external dataset used, so it is unclear how much their approach would generalize.

**Other Comments Or Suggestions:**

* I suggest to also capitalize "Reasoning" in the title.
* line 16, second column: "(CoT; Wei et al. (2023))" - wrong cite command
* line 111: "section 5" - "Section" should be capitalized to be consistent
* line 143, second column: I believe the capital R should be a lowercase r to be consistent.
* 3.1/3.2/3.3: steps s are sometimes bold and sometimes not
* line 250/251: "Algorithm~1" should be on a single line
* line 254: should be "Tables" instead of "Appendix" (or the correct Appendix A.5 should be referred to)
* 5.1: recall is sometimes used capitalized and sometimes not
* table 3, caption: "Error identification" - should not be capitalized; "Premises" -> "Model Premises" to be consistent
* line 410: "Identification :" - extra white space before the colon
* table 4, caption: "Error identification" - should not be capitalized
* table 5: unclear from the caption what data(set) is shown
* line 406/407, second column: "Table~3" - should be on a single line
* line 414/415, second column: "Table~4" - should be on a single line
* references:
  * "Alexa arena: A user-centric interactive platform for embodied ai" - at least abbreviations should be capitalized; URL cuts into the margin
  * "OpenAI o1 System Card" - cited twice
  * I suggest to cite the conference versions for example CoT (NeurIPS'22) or ToT (NeurIPS'23)
  * "Metamath: Bootstrap your own mathematical questions for large language models" - cited differently than the other arXiv papers
* line 853: should be "Tables" instead of "Appendix"
* line 1013: missing brackets around "step"

**Other Strengths And Weaknesses:**

The writing is clear and the paper is easy to follow. While the evaluation is detailed to a certain degree, for example the number of evaluated models, other aspects such as external datasets or performance of baseline models are missing.

**Questions For Authors:**

* Why do longer reasoning chains make it harder to verify individual steps?
* If the nodes are the reasoning steps, how can the edges link to premises?
* Why are there more PARCs than chains in the dataset? (A2)

**Relation To Broader Scientific Literature:**

The ideas are for the most part novel. The use of the DAG to model the dependencies between thoughts/reasoning step is somewhat akin to Graph of Thoughts, however is here applied after the generation is finished instead of during the generation, which is then used to identify errors.

**Theoretical Claims:**

Yes, to a certain degree, I have checked their formulas, which are reasonably most of the time, but contain some errors as well some notation mistakes.

* line 143, second column: I believe the capital R should be a lowercase r to be consistent.
* line 202: wrong symbol, if my understanding is correct - should be I instead of F
* algorithm 1: input and output of algorithm should be a small r?

---

> ### Author Rebuttal · Authors · 2025-04-01
>
> We thank the reviewer for their appreciation of the writing and experiments. Here we address their concerns
>
> **Typos** - Thanks for bringing those to our notice, we will definitely fix those in the final version.
>
> **concern 1** - Dataset is a bit small, no external datasets used
>
> **response** -
> To further provide proof that our method is effective we provide results on the popular Processbench dataset, which has step level annotations done by humans.
>
> Gsm8k (400 examples)
> | Model      | Method   | Correct acc | Wrong acc | F1   | Δ (Delta) |
> |------------|----------|-------------|-----------|------|-----------|
> | Llama 8B   | Baseline | 17.1        | 36.7      | 23.3 | –         |
> |            | Ours     | 33.7        | 37.8      | 35.6 | 12.3      |
> | Llama 70B  | Baseline | 77.7        | 57.5      | 66.1 | –         |
> |            | Ours     | 89.6        | 70.0      | 78.6 | 12.5      |
> | Qwen 7B    | Baseline | 66.3        | 36.7      | 47.2 | –         |
> |            | Ours     | 60.1        | 38.6      | 47.0 | -0.2      |
> | Qwen 32B   | Baseline | 97.9        | 43.0      | 59.8 | –         |
> |            | Ours     | 95.9        | 55.1      | 70.0 | 10.2      |
> | Qwen 72B   | Baseline | 98.4        | 61.4      | 75.6 | –         |
> |            | Ours     | 97.8        | 59.7      | 74.1 | -1.5      |
>
>
> Math (1000 examples)
>
> | Model      | Method   | Correct acc | Wrong acc | F1   | Δ (Delta) |
> |------------|----------|-------------|-----------|------|-----------|
> | Llama 8B   | Baseline | 5.6         | 19.1      | 8.7  | –         |
> |            | Ours     | 11.0        | 27.5      | 15.7 | 7.0       |
> | Llama 70B  | Baseline | 32.4        | 32.8      | 32.6 | –         |
> |            | Ours     | 61.6        | 55.4      | 58.3 | 25.7      |
> | Qwen 7B    | Baseline | 46.0        | 25.4      | 32.7 | –         |
> |            | Ours     | 45.6        | 41.2      | 43.3 | 10.6      |
> | Qwen 32B   | Baseline | 90.0        | 22.4      | 35.9 | –         |
> |            | Ours     | 86.9        | 53.9      | 66.5 | 30.7      |
> | Qwen 72B   | Baseline | 88.5        | 33.7      | 48.8 | –         |
> |            | Ours     | 86.7        | 53.9      | 66.5 | 17.7      |
>
>
> **concern 2** -  the baseline performance of these models is missing
>
> **response** - In our paper we have two tasks - premise identification and error identification. The former is a novel task in itself, hence we don’t explicitly compare against any baseline as such. Instead we compare the aggregative and dyadic approaches as an ablation to how premises can be identified. For the error identification task, the baseline is when the entire context is fed into the LLM as compared to our approach where we only use the premises generated (which is a standard LLM as a judge scenario, please refer to the Table 3 the rows tagged "full context" is the baseline here). We would also like to highlight that existing popular frameworks like Roscoe and Receval assign a chain level scores, and hence are not directly comparable to our use case.
>
>
> **concern 3** - discussion of the additional compute resources, time and costs
>
> **response** - As we query the judge LLM twice we observe a 2X latency as compared to the baseline. We empirically verified this for the Qwen32b model, across all 4 datasets.
>
>
> **question 1** - Why do longer reasoning chains make it harder to verify individual steps?
>
> **response** - The longer a reasoning chain gets, the longer the context is for the later steps. This further implies that a lot of irrelevant information (non-premises) is fed into the context, making it hard for the model to reason whether the step was correct or not.
>
>
> **question 2** - If the nodes are the reasoning steps, how can the edges link to premises?
>
> **response** - In our formulation in the DAG - each node is a step, and directional edges capture dependencies between steps (nodes). More specifically, if a step i is premise to step j, there is a step (i->j)
>
>
> **question 3** - Why are there more PARCs than chains in the dataset? (A2)
>
> **response** - Indeed it should be 607 and we will correct this in our final paper

---

### Official Review · Reviewer_gQAd · 2025-03-12

**Overall Recommendation:** 3

**Summary:**

This paper studied the step-level verification of CoT reasoning, and proposed a PARC framework that converted linear reasoning chain into DAG by introducing premise links. Based on the framework, the authors defined a new error type named accumulation error, and constructed PERL dataset to evaluate the framework. The experiments demonstrated that PARC helped to identify step-level errors in CoTs.
## update after rebuttal
The responses have addressed most of my concerns. After reading other reviews and authors' responses, I decide to keep my score.

**Claims And Evidence:**

The claims are well supported by the experimental results. The authors specially constructed several datasets and conducted extensive experiments to demonstrate the effectiveness of the proposed framework, including the premise identification and error identification.

**Essential References Not Discussed:**

Related works are well cited.

**Experimental Designs Or Analyses:**

The experiments are extensive and solid. I have one concern that in dataset construction, it may be problematic to treat the CoT as correct based on the final answer, due to possible step errors as studied in the paper.

**Methods And Evaluation Criteria:**

Overall the proposed method works for CoT verification. However, there are still some issues that are not clearly explained in method design.
1. Since the premise is based on the step, it would be better to make it clearer what is an intermediate step defined, and how to get the step list from a complete CoT.
2. How did the authors ensure or evaluate whether the premise satisfied the three properties in page 3. And how to evaluate whether a step is verifiable.
3. What’s the necessity of the proposed accumulation error. According to the paper, if the error exists, it means that there are native errors, which is enough to indicate the incorrect reasoning.

**Other Comments Or Suggestions:**

None.

**Other Strengths And Weaknesses:**

In summary, the strengths of the paper are listed as follows.

1. The paper proposed a novel premise DAG-based method PARC for step-level verification of CoT reasoning.
2. The authors conducted extensive experiments to demonstrate the effectiveness of the framework.

The weakness of the paper are as follows.

1. Some details of method design are unclear in the paper. See the "Methods" and "Experiments" parts.

**Questions For Authors:**

1. How did the authors ensure or evaluate whether the premise satisfied the three properties in page 3. And how to evaluate whether a step is verifiable.
2. What’s the necessity of the proposed accumulation error. As the native errors are enough to indicate the incorrect reasoning.

**Relation To Broader Scientific Literature:**

The paper focused on the verification of the CoT based on premise DAG. Compared with existing works, the authors proposed an automatic premise recognition method without relying on predefined CoT template and hurting reasoning, and propose a novel error type focusing on error propagation.

**Theoretical Claims:**

There is no theoretical claim in the paper.

---

> ### Author Rebuttal · Authors · 2025-04-01
>
> We sincerely thank the reviewer for their kind remarks on the novelty and thorough experimental setup of our method. Here we try to address their concerns.
>
> **concern 1** - Since the premise is based on the step, it would be better to make it clearer what is an intermediate step defined, and how to get the step list from a complete CoT.
>
> **response** -  In our work, we explicitly prompt the generator model to answer in a step by step format, with formatting instructions in the prompt. The generated solution looks like
> Step 1: …
> Step 2: …
> Finally we do a simple regex to extract the steps.
>
> **concern 2** - How did the authors ensure or evaluate whether the premise satisfied the three properties in page 3.
>
> **response** - In our work, we prompt the highly capable O1-preview model to generate the ground truth premises and later manually verify the generated premises on whether they satisfy the verifiability and minimality conditions.
>
>
> **concern 3** - What’s the necessity of the proposed accumulation error?
>
> **response** - A holistic evaluation of LLM reasoning should consider the entire reasoning chain rather than relying solely on a binary correct/incorrect outcome of the final answer. Reasoning chains may contain subtle intermediate errors despite following a globally correct plan, ultimately rendering the solution incorrect, a nuance overlooked by final-answer correctness metrics. Prior works like PRM800K and ProcessBench have typically annotated reasoning chains only up to the first erroneous step, discarding subsequent steps due to ambiguity. To our knowledge, we are the first to formally introduce the concept of accumulation errors, enabling a more comprehensive evaluation of reasoning chains. Similar to how teachers award partial credit for nearly correct answers, evaluation frameworks should recognize when the overall reasoning plan is sound despite minor mistakes, assigning partial credit accordingly. Accumulation errors, where a step is locally correct but built on flawed premises, explicitly capture this subtlety. Identifying accumulation errors highlights how earlier mistakes compromise the reliability of the reasoning chain, making it essential to incorporate these errors into holistic scoring methods.
>
> **concern 4** - it may be problematic to treat the CoT as correct based on the final answer, due to possible step errors as studied in the paper.
>
> **response** -
> We completely agree with the reviewer on this - in our work, we observed that even in the positive reasoning chains, there were a few steps that were annotated as error by the O1 model, indeed LLMs suffer from False Positives, which is established from the work - https://arxiv.org/pdf/2502.06217 .
> In GSM8k - 4 false positives out of 50
> In MATH - 4 out of 50
> Metamathqa - 5 out of 50
> Orcamath - 4 out of 50
> When we release our dataset, we will make sure to flag them as false positives.

---

> > ### Comment · Reviewer_gQAd · 2025-04-04
> >
> > Thanks for the time on responses. The responses have addressed most of my concerns. According to the responses, the proposed method can not be applied on more general CoTs, and the authors have not technically ensured the satisfcation of premise property, so I would keep my score unchanged.

---

> > > ### Author Response · Authors · 2025-04-04
> > >
> > > We thank the reviewer for their time and feedback. Here is our response to their concerns
> > >
> > > 1. **proposed method can not be applied on more general CoTs**
> > >
> > > We would like to highlight that we also did experiments on the processbench dataset (the results are available on **concern 1** by **reviewer kjs7**), where the chain of thoughts are **not** generated in a step by step manner and a simple delimiter is used to separate steps. And our method consistently outperforms the baseline in that case as well. Our experiments show that PARC consistently outperforms verification baselines irrespective of the style in which the reasoning chain was generated.
> > >
> > > 2. **authors have not technically ensured the satisfaction of premise property**
> > > Unfortunately reasoning chains are in natural language and hence highly ambiguous. Which is why we annotate a high quality dataset against which predicted premises can be compared. Here is how we ensured the satisfaction of the premise properties
> > >
> > >     **For our PERL dataset:** We ensure that both the premise properties (verifiability and minimality) are satisfied for our PERL dataset. At the time of construction of the dataset, we manually check the generated data to make sure the conditions are met.(detailed in lines 314-328)
> > >
> > >     **At inference time:** At inference time we compare the premises generated by the models with the ground truth premises to report our accuracy metrics. Since we already have a reliable set of premises, these metrics capture well how good the models are in predicting premises.

---

### Official Review · Reviewer_XAi3 · 2025-03-15

**Overall Recommendation:** 3

**Summary:**

This paper introduces a new category of errors (accumulation errors) and Premise-Augmented Reasoning Chains (PARC) as a method to improve error identification in mathematical reasoning with Large Language Models (LLMs). To evaluate this method, the authors construct PERL (Premises and ERrors identification in Language models), a benchmark dataset containing annotated premise links and error types in mathematical reasoning chains. Their results demonstrate that LLMs can achieve ≥90% recall in premise extraction, highlighting the effectiveness of their method.


## update after rebuttal
I'd like to thank the authors for their response and keep my positive rating.

**Claims And Evidence:**

The authors claim that “off-the-shelf LLMs can detect premises for a given step with high accuracy for mathematical reasoning.” However, my main concern lies in the low precision of premise identification and mapping, as reported in Tables 1 and 2. While I acknowledge the high recall achieved by LLMs, precision remains a critical factor, particularly given the minimality requirement defined in Lines 151–164—where a premise set should be minimal such that removing any element renders the corresponding step unverifiable. With precision ranging from 60% to 80%, the identified premises do not seem to consistently meet this minimality criterion, leaving the authors' claim insufficiently substantiated.

**Essential References Not Discussed:**

Graph-of-Thoughts [1] also discusses non-linear reasoning structures, which relate to PARC’s DAG-based structure but is not cited. A discussion would be helpful.

[1] Besta, Maciej, et al. "Graph of thoughts: Solving elaborate problems with large language models." Proceedings of the AAAI Conference on Artificial Intelligence. Vol. 38. No. 16. 2024.

**Experimental Designs Or Analyses:**

The paper presents a thorough experimental design, with evaluations conducted across multiple datasets such as GSM8K and MATH, ensuring a diverse assessment of the proposed method. However, a major concern lies in the experimental setup described in Section 5.2, particularly in Lines 373–378, where the authors state that Mathematical Error and Logical Inconsistency are merged into a single error type, Error, due to their “thin boundary”. Since these are well-established and distinct error types, the inability of models to differentiate between them suggests a limitation in model capacity rather than an inherent issue with error categorization. Merging these categories appears to artificially simplify the task, potentially leading to an ad-hoc and unfair evaluation of performance. A more rigorous analysis preserving the original error distinctions would provide a clearer assessment of the model’s reasoning capabilities.

**Methods And Evaluation Criteria:**

The PARC framework is well-motivated and introduces an intuitive premise-based verification process for mathematical reasoning. The PERL dataset encompasses a diverse set of math word problems from GSM8K, MATH, Orca-Math, and MetaMathQA, ensuring a broad range of difficulty levels. Additionally, the chosen evaluation metrics—precision, recall, and F1 for premise detection, along with accuracy for error identification—are appropriate for assessing the effectiveness of the proposed approach.

**Other Comments Or Suggestions:**

See previous sections.

**Other Strengths And Weaknesses:**

See previous sections.

**Questions For Authors:**

See previous sections.

**Relation To Broader Scientific Literature:**

This work introduces a new error taxonomy by defining accumulation errors and situates itself within existing research on reasoning verification. While building on prior work, it presents a novel structured verification and error detection approach, contributing a more systematic method for identifying and analyzing reasoning errors in math reasoning tasks.

**Theoretical Claims:**

The formalization of premise extraction and mapping problem (Section 3.1) is reasonable. However, the paper lacks in-depth theoretical guarantees or analysis on the solution quality of the proposed algorithm (Algorithm 1), leaving open questions about its optimality and robustness in premise identification.

---

> ### Author Rebuttal · Authors · 2025-04-01
>
> We thank the reviewer for their kind remarks on the novelty and thorough experimental design. Here we address the concerns
>
> **concern 1** - low precision of premise identification and mapping
>
> **response** - We would like to highlight that our claim is originated by the high recall primarily. In the context of error identification, verifiability (recall) is more important than minimality (precision), since missing even a single premise can compromise the step’s and subsequent steps' verifiability. Further, upon closer inspection we observe that, low precision often results from the ground truth premise set being very small. For example, if step 7 has ground truth premises {1} and the model predicts {0,1}, precision is 50%, even though the context remains well-pruned.
>
> **concern 2** - lack of theoretical guarantees
>
> **response** - We primarily take an empirical route, and as already pointed out by you, we do thorough experiments across model families, scale and datasets to prove robustness of our method. We hope that the released dataset can help the community to  foster more research in this direction as well.
> To further provide proof that our method is effective we provide results on the popular Processbench dataset, which has step level annotations done by humans. **Please refer to our response to concern 1 raised by Reviewer kjS7 for the Processbench results.**
>
>
> **concern 3** -  Mathematical Error and Logical Inconsistency are merged
>
> **response** - When we say that the boundary between them is “thin”, we imply that even a simple mathematical calculation (or manipulation) could easily be considered as a logical error (due to fault in mathematical logic). However we agree that for completeness, it's essential to have a detailed breakdown per error category, which we present here. ( we will include these in the final version of the paper)
>
>
> GSM8k
> | Model            | Context        | Logical Error | Mathematical Error |
> |------------------|----------------|---------------|--------------------|
> | Llama 3.1 8b     | Full context   | 29.9          | 41.1               |
> |                  | Model premises | 55.6          | 64.9               |
> | Llama 3.1 70b    | Full context   | 48.5          | 72.4               |
> |                  | Model premises | 79.6          | 64.5               |
> | GPT4o-mini       | Full context   | 33.4          | 58.6               |
> |                  | Model premises | 64.7          | 65.8               |
> | GPT-4o           | Full context   | 44.4          | 52.4               |
> |                  | Model premises | 64.1          | 74.0               |
> | Qwen 7b          | Full context   | 21.4          | 20.7               |
> |                  | Model premises | 59.0          | 39.1               |
> | Qwen 72b         | Full context   | 31.7          | 44.5               |
> |                  | Model premises | 69.9          | 64.6               |
>
>
> MATH
>
> | Model            | Context        | Logical Error | Mathematical Error |
> |------------------|----------------|---------------|--------------------|
> | Llama 3.1 8b     | Full context   | 46.4          | 45.1               |
> |                  | Model premises | 57.9          | 63.7               |
> | Llama 3.1 70b    | Full context   | 46.0          | 75.0               |
> |                  | Model premises | 79.0          | 66.3               |
> | GPT4o-mini       | Full context   | 54.1          | 61.5               |
> |                  | Model premises | 83.1          | 67.4               |
> | GPT-4o           | Full context   | 55.1          | 59.4               |
> |                  | Model premises | 63.4          | 64.2               |
> | qwen 7b          | Full context   | 27.7          | 34.7               |
> |                  | Model premises | 77.6          | 49.2               |
> | qwen 72b         | Full context   | 47.8          | 63.8               |
> |                  | Model premises | 77.7          | 66.8               |
>
>
>
> **concern 4** - Graph-of-Thoughts [1] also disc…
>
> **response** - Thanks for raising this. Graph of thought induces structure at inference time, while we do it after inference, and later use that to improve error identification. But there is definitely a resemblance between them, and we will include them in our camera ready version.

---

### Decision · Program_Chairs · 2025-05-01

**Decision:**

Accept (poster)

**Comment:**

The paper presents a method to construct premise-augmented reasoning chains, which are useful in increasing the verification capabilities of language models. It also provides a way to quantify the accumulation kind of error in the target model's reasoning. Reviewers appreciated the motivation of the framework and its application to math benchmarks. The addition of another benchmark, ProcessBench during the rebuttal was useful.
Overall, the paper presents a useful idea to improve the capability of LLMs to evaluate reasoning.